**Perspective**

# Harnessing health economic evaluation for policy and practice

Yot Teerawattananon [1,2], Yi Wang [2 ✉], Sarin KC[1], Hugo C. Turner[3], Benjamin SK Ong[4] & Wanrudee Isaranuwatchai[1]

Health economic evaluation is essential for evidence-informed health policy, supporting prioritization of high-value care and optimal resource use. This Perspective synthesizes experiences from the HTAsiaLink network and global literature to demonstrate its role across the technology lifecycle. Drawing on diverse country experiences, it highlights how these methods guide value-based, equitable, and sustainable decision-making amid rapid technological advances and resources constraints. Key challenges and opportunities are examined, such as integrating equity considerations, incorporation of environmental impacts, and adapting frameworks for novel technologies. Strengthening deliberative processes, building capacity, and adopting flexible methods are crucial to harnessing full potential of health economic evaluation.

In contemporary healthcare systems, escalating costs and constrained resources underscore the necessity for transparent and systematic approaches to health prioritization and resource allocation[1]. Evidence-based decision-making is critical to maximize the efficiency and effectiveness of healthcare delivery, improve population health outcomes, and ensure system sustainability. Structured frameworks can enable policymakers to make informed, equitable, and accountable decisions aligned with societal values and health system objectives[2].

Health economic evaluations have emerged as essential methodological tools. They involve a comparative assessment of the value for money of healthcare interventions by comparing their associated costs with outcomes, frequently quantified through metrics such as quality-adjusted life years (QALYs) or disability-adjusted life years (DALYs)[3]. These evaluations support the identification of high-value healthcare options, enabling prioritization that maximize health benefits within constrained budgets.

The rapid pace of technological innovation further underscores the need for health economic evaluation and evidence[4]. New therapies and diagnostics introduce challenges related to cost, equity, and value[5]. Health economic evaluations can help direct resources toward cost-effective interventions, mitigate inefficiencies, and reduce unwarranted care variations. Their adoption is increasing globally[6], reflecting a growing recognition of their role in supporting universal health coverage (UHC) and efficient, equitable service delivery[7]. However, application varies widely based on institutional capacity, methodological standards, and political contexts.

This Perspective offers a comprehensive overview of how health economic evaluations have been used to inform health policy, synthesizing experiences from diverse high-income countries and low- and middle-income countries (LMICs). The core analysis draws on the practical experiences and shared learnings of the HTAsiaLink network, a collaboration of national Health Technology Assessment (HTA) agencies from over 15 jurisdictions across Asia and the Pacific[8]. To broaden the scope and ensure global relevance, this foundation is supplemented with a review of key literature from seminal studies and major HTA institutions worldwide.

Through this examination of varied contexts, the paper identifies cross-cutting best practices, common challenges, and critical lessons for integrating economic evidence throughout the technology lifecycle, from innovation to disinvestment. By combining regional insights with international perspectives, this Perspective provides a nuanced and practical foundation for policymakers and stakeholders seeking to strengthen health economic deliberation and promote more sustainable, equitable, and efficient health systems.

## The role of health economic evaluation across the healthcare technology lifecycle

This section illustrates the application of health economic evaluations through real-world case studies, highlighting how they balance clinical effectiveness with economic sustainability across the healthcare technology lifecycle. Health economic evaluations provide critical insights, such as comparative effectiveness and value for money to assess emerging innovations, as well as evidence on cost-effectiveness and budget impact to inform pricing, guideline development, and reevaluation decisions that may guide disinvestment, thereby supporting efficient and sustainable healthcare delivery.

[1]Health Intervention and Technology Assessment Program Foundation, Ministry of Public Health, Nonthaburi, Thailand. [2]Saw Swee Hock School of Public Health, National University of Singapore and National University Health System, Singapore, Singapore. [3]MRC Centre for Global Infectious Disease Analysis, School of Public Health, Imperial College London, London, UK. [4]Agency for Care Effectiveness, Ministry of Health, Singapore, Singapore. ✉e-mail: ephwyi@nus.edu.sg

### Early health economic evaluation

Early health technology assessment (eHTA) is a relatively new approach that is not yet widely adopted but holds significant promise in health innovation development[9–12]. Within the broader eHTA framework, early health economic evaluations can be used to assess the potential value for money and feasibility of new interventions. These evaluations often involve engaging potential patients and users, providing developers with valuable insights into their needs, gaps, preferences, and real-world considerations. Unlike traditional health economic assessments that often focus on calculating incremental cost-effectiveness ratios (ICERs) for fully specified interventions, early evaluations aim primarily to identify key intervention characteristics and usage pathways that can position the intervention's ICER just below or at the relevant cost-effectiveness threshold (CET) within specific settings. By reverse-engineering traditional health economic evaluation, early health economic evaluation provides forward-looking insights on how technology can become cost-effective and adopted.

The information gathered from early health economic evaluation is instrumental not only for defining target product profiles (TPPs) but also for guiding the design of the intervention's application within healthcare systems and shaping clinical study development[13]. Early insights generated through this process can inform future clinical studies needed to demonstrate safety and efficacy for market authorization, as well as support post-market health economic evaluations for coverage decisions, ultimately accelerating the pathway to market approval and reimbursement.

A survey among HTAsiaLink members identified one research team, Medical Innovation Development and Assessment Support (MIDAS), under Thailand's Health Intervention and Technology Assessment Program Foundation (HITAP) and the National University of Singapore, that focuses on eHTA in collaboration with health innovators and health innovation funders from both the public and private sectors. Box 1 presents an example of early HTA work from the MIDAS team. Early health economic evaluation was performed in conjunction with a landscape review and stakeholder consultations.

Two other related concepts are early dialogue (also known as early scientific advice) and horizon scanning. Scholars hold differing views on whether these two should be considered components of eHTA methods or stand-alone activities. Early dialogue refers to exchanges between private sector innovators and public institutions to obtain guidance on evidence requirements for regulatory and reimbursement purposes[14]. Horizon scanning, on the other hand, is the systematic identification of health technologies that are new, emerging, or becoming obsolete, and that may significantly affect health, healthcare services, or society[15]. Horizon scanning identifies, filters, and prioritises new and emerging health technologies, or new uses of existing interventions, and assesses their potential impact on health and healthcare system before they diffuse locally. When available, economic evidence is collected but typically summarized qualitatively due to limited data. This information helps policymakers and healthcare institutions better prepare for adoption and integration of innovative technologies.

### Reimbursement decisions and price negotiation

The use of health economic evidence to inform reimbursement and coverage decisions for new health interventions has been widely adopted across many countries in the Asia-Pacific region over the past several decades. For example, Australia has utilized health economic assessments through the Pharmaceutical Benefits Advisory Committee (PBAC) for coverage decisions of pharmaceuticals over the past 35 years[16]. Similarly, the Medical Devices and Human Tissue Advisory Committee (MDHTAC) has incorporated health economic evaluations for coverage decisions, establishing a robust framework for integrating economic evidence into policy[17]. More recently, countries such as the Philippines have formally adopted health economic evaluation as a mandatory component of their UHC process, with its use becoming standard practice over the past few years[18]. The Philippines' UHC law also has a clause to highlight the importance of HTA. This approach helps policymakers assess whether a new intervention provides sufficient value relative to its costs, especially when compared to existing standards of care. Often, health economic evaluations are conducted alongside budget impact analyses, enabling decision-makers to make more informed choices about which technologies should be included in publicly funded programs or insurance schemes[19].

ICER, which compares the additional costs and health benefits of a new intervention relative to a comparator, together with the CET, provides the basis for assessing value for money[3]. However, how countries operationalise these concepts in price negotiations varies considerably. In the Philippines and Australia, HTA informs reimbursement without a formally applied CET: the Philippines uses assessments of Health Technology Assessment Council to support a structured but relatively new price negotiation process, while Australia relies on implicit benchmarks derived from past PBAC decisions and employs managed entry agreements for uncertain or high-cost technologies[16,18]. Managed entry agreements can generally be classified into two types: financial-based and outcome-based[20,21]. Financial-based agreements improve patient access while controlling costs by limiting reimbursed volume or reimbursement levels and sharing expenditures with manufacturers. Outcome-based agreements adjust and share payments according to the real-world performance of the therapies. Financial-based managed entry agreement has been used more often in Australia[22]. In contrast, Thailand and Japan apply explicit CETs more directly as tools for price intervention. Thailand's Price Negotiation Working Group uses guideline-recommended price-threshold analyses, by calculating the

---

## Box 1 | Early health technology assessment of tongue swab for non-sputum based pulmonary tuberculosis diagnosis in Thailand[79]

In collaboration with a group of innovators developing tongue-swab-based essays for pulmonary tuberculosis (PTB) diagnosis, the MIDAS team conducted early health technology assessment to explore the potential role and value-for-money of tongue-swab sample collection for PTB detection within Thailand's healthcare system. Following a scoping review of tuberculosis (TB) diagnosis, screening, and care pathways in Thailand, a focus group discussion was conducted with stakeholders including TB clinicians, pediatric specialists, researchers, and policy-makers from Thailand's Division of Tuberculosis, and the innovators. An early health economic evaluation was then performed using the input from literature and expert opinions. Tongue swab with real-time polymerase chain reaction (RT-PCR) and tongue swab with Loop-Mediated Isothermal Amplification (LAMP) were compared with the current practices, including acid-fast bacillus smear microscopy with sputum Xpert testing for individuals aged above 5 years and tuberculin skin test for children under age 5 years. Using the Thai CET 160,000 Tahi Baht per QALY[80], tongue swab with RT-PCR were found to be cost-effective for individuals aged above 5, but tongue swab with LAMP was not. For children under 5 years, both tongue swab with RT-PCR and tongue swab with LAMP were cost-effective. TPP characteristics including sensitivity, specificity, cost of the innovation, and test and treatment non-compliance rate were identified. Through this collaboration, the innovators obtained insights to refine their technology, and the innovation was introduced to potential stakeholders and end-users for future engagement.

## Box 2 | Health economic evaluation of intervention thresholds for patients with osteopenia: the ACE Clinical Guideline in Singapore

Individuals with fragility fractures or bone mineral density (BMD) T-scores ≤ −2.5 receive an osteoporosis diagnosis and commence pharmacological treatment, as these factors significantly increase fracture risk. Early intervention in these patients prevents future fractures and associated morbidity. For individuals with BMD between −1.0 and −2.5, however, the balance between treatment benefits and potential harms becomes more nuanced. Treatment decisions for this group should therefore be guided by overall fracture risk assessment rather than BMD measurements alone. This risk-based approach enables more precise identification of patients likely to benefit from intervention.

The ACE Clinical Guideline on osteoporosis in Singapore recommended locally developed thresholds derived through cost-

effectiveness modelling to guide treatment initiation in osteopenic patients. The decision to employ health economic evaluation was based on the guideline's population-level perspective and the need for country-specific thresholds that reflect Singapore's unique fracture epidemiology and healthcare costs. The analysis used a Markov model that simulated fractures and mortality across age- and sex-specific cohorts. This approach identified the fracture risk threshold at which treatment became cost-effective compared to no intervention. Following rigorous deliberations and consensus methods by the guideline panel, these evaluation findings informed the strength of treatment recommendations for osteopenic patients.

maximum price at which a technology would be cost-effective, to negotiate with manufacturers, achieving substantial budget savings[23]. Japan has introduced formal ICER-based post-listing price adjustments, where technologies exceeding specified ICER thresholds may undergo price reductions upon reassessment[24].

A persistent challenge for health systems is the "cost-effective but unaffordable" paradox: an intervention may fall below a cost-effectiveness threshold yet still generate a budget impact that threatens fiscal sustainability. Countries have adopted various strategies to manage this tension, such as staggered implementation for hepatitis C treatment[25] and managed entry agreements for high-cost medicines such as imiglucerase[26], which cap volumes or share financial risk with manufacturers. However, a core principle of HTA is often misunderstood. HTA does not imply that all cost-effective interventions must be adopted. Rather, it is a priority-setting tool that enables decision-makers to say "no" to technologies that do not represent good value for money or are unaffordable, thereby safeguarding financial sustainability and avoiding the crowding out of other priority services. Evidence from Thailand illustrates this clearly: nearly one-third of proposed cancer medicines were never formally submitted for reimbursement because they were anticipated to be unaffordable, reflecting a form of pre-emptive rationing based on expected budget impact[27]. This highlights that affordability considerations may, at times, need to override cost-effectiveness alone, and that transparent criteria and explicit trade-off decisions are essential to ensure limited resources maximise population health gains.

Taken together, the experience from Asia-Pacific region highlights a pragmatic use of CETs as operational levers for price negotiation, contrasting with European countries and Canada where thresholds more commonly serve as reference benchmarks for coverage decisions rather than explicit tools for price adjustment[28]. This process, while recognising other decision criteria (e.g. unmet need, system readiness), promotes sustainable resource allocation and facilitates access to cost-effective innovations.

### Clinical practice guideline development

Incorporating economic evidence into clinical practice guidelines ensures that recommendations are grounded not only in terms of clinical effectiveness but also in cost-effectiveness[29]. This approach helps balance the benefits of healthcare interventions with their associated costs, guiding clinicians and policymakers toward practices that maximize patient outcomes while promoting sustainable use of resources. By analyzing the economic impact alongside clinical data, healthcare systems can prioritize interventions that offer the greatest value for money, reduce unnecessary expenditures, and enhance overall efficiency.

However, this strategy faces certain challenges. A primary limitation is that health economic evaluations often focus on individual technologies or

specific interventions rather than the entire process of care. This approach means that the analysis needs to consider the comprehensive care pathway or treatment process, rather than isolated procedures or devices. Consequently, applying economic evidence to inform guidelines requires a thorough understanding of the full care process, which can be complex and resource-intensive to develop. This contributes to the persistently low rate of adoption of economic evidence in guidelines, its frequent use as a superficial, post-hoc justification rather than an integral methodological component, and the lack of rigor in identifying and appraising health economic studies[30].

Despite these difficulties, integrating economic considerations remains vital for developing guidelines that are both scientifically robust and economically sustainable, ultimately supporting the adoption of high-value practices that improve access, optimize health outcomes, and ensure responsible resource utilization across populations. Box 2 presents an example of this application by the Agency for Care Effectiveness (ACE), Singapore's national HTA and clinical guidance agency.

### Reevaluation and disinvestment

As evidence continues to evolve, certain healthcare interventions may become outdated, demonstrate limited effectiveness (than originally expected), or offer marginal value. Health economic evaluations play a critical role in identifying such low-value practices by systematically assessing their cost-effectiveness and their impact on patient outcomes[31]. This evidence base underpins disinvestment strategies, enabling healthcare systems to redirect resources from interventions that provide minimal benefit toward those that deliver greater value, thereby optimizing health outcomes and system efficiency.

From a technical perspective, the process of disinvestment is relatively straightforward, as it primarily entails monitoring the invested intervention, reviewing existing evidence and revising funding priorities. Politically, however, it remains considerably more complex[32]. Disinvestment often encounters resistance from clinicians, patients, industry stakeholders, and policymakers, many of whom may have vested interests, emotional attachments, or concerns about perceived losses associated with the withdrawal of support for particular interventions. Moreover, cultural and organizational barriers, including apprehensions about public perception, access, and equity, can further complicate decision-making. Addressing these challenges requires transparent communication, inclusive stakeholder engagement, and strong leadership committed to advancing value-based care. Ultimately, the systematic integration of evidence-based disinvestment strategies enhances resource efficiency, supports sustainable healthcare delivery, and strengthens overall system performance.

As development assistance for health faces mounting fiscal pressures and widespread funding cuts, the imperative for evidence-based disinvestment becomes even more acute. In this constrained environment, health

## Box 3 | Using health economic evaluation to delist bevacizumab in Indonesia[81]

The case of bevacizumab for metastatic colorectal cancer (mCRC) in Indonesia exemplifies the political dynamics of health technology reassessment and subsequently disinvestment under UHC. A health economic evaluation for the national payer, BPJS Kesehatan, established that bevacizumab was not cost-effective. The study found that the drug offered only a minimal clinical benefit, such as approximately one month of longer life, at a very high cost, posing a significant threat to the financial sustainability of the Jaminan Kesehatan Nasional (JKN) scheme. This evidence-based finding aligned with international standards. For example, Thailand's UHC scheme does not reimburse bevacizumab for mCRC, and the UK's National Institute for Health and Care Excellence

also does not recommend it due to its poor value for money[82]. In contrast, Indonesia's JKN benefits package was initially more generous.

Based on this evidence, the Minister of Health formally delisted the drug from the national formulary, a decision projected to yield annual savings of US$14 million. However, this ministerial decision was subsequently overturned following intervention by the national parliament, which responded to intense advocacy from clinical societies and patient groups. This sequence reveals a critical implementation gap: technically justified decisions require not only robust evidence but also a strategy to build consensus across government branches, highlighting the fundamental tension between technocratic management and democratic political processes in health financing.

---

systems must make difficult trade-offs, ensuring that limited resources are directed toward interventions that deliver the greatest impact over time, not only at the first assessment. Box 3 presents an example from Indonesia on delisting of bevacizemab.

### Other applications

Beyond their core functions, health economic evaluations exert wide-ranging influence on health policy and practice, supporting more evidence-informed and effective decision-making across the healthcare system. An important application lies in guiding professional education. By providing evidence-based information on the relative benefits, risks, and costs of alternative interventions, health economic evaluations equip healthcare providers and patients to make choices that align with the principles of value-based care. This evidence base fosters shared decision-making, encourages the uptake of efficient and effective practices, and strengthens patient-centered approaches to care. However, a knowledge gap impedes this application; many physicians lack proficiency in health economic methods, limiting the use of evaluations in practice[33]. Targeted education is therefore critical, as evidence shows that formal training significantly increases both understanding and the adoption of cost-effectiveness principles.

Health economic evaluations also contribute to the design and implementation of performance-based frameworks such as the Quality and Outcomes Framework, which links financial incentives to measurable health outcomes[34]. Within this context, health economic evaluations inform the development of incentive structures that encourage healthcare providers to prioritize interventions delivering the greatest health benefits relative to resource use. A key challenge is the reliance on activity-based or surrogate outcomes for commissioning and payment, which can diverge from health economic evaluation evidence that emphasizes final outcomes like QALYs[35]. A Good Practices Report on methods for evaluation of surrogate endpoints for HTA decision making emphasizes validating surrogates at three levels: 1) treatment effect association (does treatment change in the surrogate predict change in the final outcome?); 2) outcome association (are surrogate and final outcome associated?); and 3) biological plausibility. Recommended methods include multivariate meta-analysis, which accounts for differences between studies and correlations within them; Bayesian approaches, which help strengthen conclusions when data are limited; and the use of real-world data, which can complement trial evidence, provided that potential biases are carefully assessed and managed[36]. An example could be when an individual participant data (IPD) meta-analysis was conducted to assess surrogacy, showing that disease-free survival is a strong surrogate for overall survival in HER2-positive breast cancer patients treated with trastuzumab[37]. However, when Canada's Drug Agency (CDA) assessed pertuzumab with trastuzumab and taxane for neoadjuvant treatment of HER2-positive breast cancer, an IPD meta-analysis found that pathological complete response did not reliably predict overall survival[38]. Due to this weak association, the drug was not recommended.

Health economic evaluations bridge this gap by modeling the link between incentivized activities and health outcomes, ensuring surrogates are valid indicators of efficiency[39]. When aligned, such frameworks discourage low-value care, improve resource allocation, and ensure rewards reflect real health gains rather than service volume, making payment structures both practical and equitable in advancing population health. Health economic evaluations also inform long-term resource planning and health system strengthening[40]. By projecting costs and benefits over extended time horizons, they enable policymakers to anticipate evolving population health needs, allocate resources efficiently, and design sustainable delivery strategies. To fully capture the comprehensive value of health systems, these evaluations must adopt innovative methodologies that account for broader impacts, including economies of scale, scope, and system-wide gains, thereby moving beyond restrictive timeframes and limited outcome measures. This evolution is essential to accurately forecast future requirements and foster the development of resilient, sustainable health systems.

### Key challenges and future directions for health economic evaluation in policy and practice

#### Broadening economic evidence in HTA framework and process

Health economic evaluations provide crucial insights but often face challenges in directly informing policy. Their focus on efficiency may overlook ethical principles, social values, and practical feasibility constraints. As a result, evidence can misalign with broader societal goals, leading to policies that are cost-effective yet ethically contentious, socially inequitable, or practically unworkable[41]. Furthermore, the transition from evidence to policy is complicated by political pressures and competing stakeholder interests, with industry groups, advocacy organizations, and political agendas potentially undermining the objectivity and legitimacy of decision-making.

Addressing these challenges requires a structured framework for synthesis and deliberation, which HTA provides[42]. Future HTA practice will need to move beyond a purely technical review toward a participatory and transparent process. This updated process could include embedding deliberative mechanisms that enable structured engagement with all stakeholders, systematically incorporating patient voices, and prioritizing transparency to counter political and stakeholder pressures. By embracing these trends, HTA can continue to develop as a vital deliberative framework, balancing economic evidence with societal values to support policies that are not only efficient but also legitimate, equitable, and widely accepted. Crucially, this comprehensive approach highlights that cost-effectiveness represents only one component of a much broader evaluative process.

#### Efficiency or equity or both? Operationalizing equity-informed analysis

A central challenge in the use of health economic evaluation for policymaking lies in its conventional emphasis on aggregate efficiency which

maximizes total health benefits but frequently overlooks equity implications. Standard analysis tends to favour interventions with the lowest cost per QALY, which may unintentionally introduce or exacerbate health inequities gap by prioritising populations that are less costly or simpler to treat[43]. By focusing predominantly on efficiency, this approach can neglect the ethical obligation to direct resources toward underserved groups, potentially resulting in policies that enhance overall health outcomes while marginalising the most vulnerable or those with unmet needs. Currently, some jurisdictions consider equity in the decision-making process, but often in a qualitative and less structured manner.

Looking forward, operationalizing equity in economic analysis necessitates a dual approach: advancing robust methodologies and building consensus on their application[44]. Commonly known methods are Extended Cost-Effectiveness Analysis (ECEA) and Distributional Cost-Effectiveness Analysis (DCEA). ECEA considers impact of health interventions on health including financial risk protection and equity, whereas DCEA focuses on how the costs and outcomes in each intervention are distributed across equity-relevant groups (e.g. income quintiles)[45,46]. The DCEA framework has been introduced to explicitly integrate distributional concerns, enabling a nuanced understanding of equity-efficiency trade-offs. By incorporating both the magnitude and distribution of health benefits, it supports more transparent and informed policy decisions that balance overall health improvement with fairness considerations. A foundational base for this work is already being established, as evidenced by pioneering group efforts within the HTAsiaLink network by countries such as Australia, India, Japan, and Thailand to create the Equity in Asia Pacific Health Technology Assessment Network (EquitAP-HTA). The aims of the network are to facilitate networking and foster a sense of community and shared purpose, share knowledge, support capacity building, and initiate collaborative research for people interested in equity in HTA. Examples of our collaborative work include sharing the results of how each country values equity through the survey on health inequality, i.e. the preference for reducing health gaps between the better-off and worse-off, a first step to estimate equity index which can be used in DCEA[47–50].

Nevertheless, the widespread implementation of economic methods which incorporate equity still faces considerable practical difficulties, especially in LMICs, where this approach demands significantly more disaggregated data and equity-relevant parameters that are often scarce or lack supporting data. Therefore, the path forward must build on existing examples by simultaneously refining the methodological framework and actively strengthening health information systems including building local technical capacity to overcome data constraints. Ultimately, the goal is to embed this method into formal decision-making criteria, creating a structured process where interventions delivering greater benefits to underserved populations can be appropriately valued, leading to more effective and equitable health policy worldwide.

Beyond distributional analyses, an increasingly adopted approach involves modifying CETs based on disease severity, reflecting societal preferences to prioritise those with greater illness burden[38]. The Netherlands applies higher thresholds (up to €80,000 per QALY) based on proportional disease burden, while the UK applies elevated thresholds for end-of-life and highly specialised technologies[51,52]. However, implementing severity-based adjustments presents significant challenges, including varying definitions of severity, limited empirical basis for threshold magnitudes, and potential opportunity costs of redirecting resources from broader population health gains[52]. Thailand's experience with Imiglucerase for Gaucher disease illustrates these tensions: despite an ICER exceeding nine times the prevailing threshold, the drug was approved for approximately five children annually[23]. Applying the value attribution framework developed by Ochalek et al.[53], this decision resulted in negative realised population net health effects, meaning health gains for treated children were outweighed by health losses elsewhere in the system due to opportunity costs. Notably, subsequent market entry of a therapeutic competitor reduced Imiglucerase's price by over 25%, demonstrating that fostering competition, rather than simply raising thresholds, can improve both access and population health. Crucially, evidence from Thailand confirms that raising thresholds without corresponding price regulation increases manufacturer share without improving population health, underscoring that severity modifications should be grounded in domestic evidence on both societal preferences and opportunity costs, and implemented alongside complementary policies such as managed entry agreements and competition-enhancing measures[53].

## Integrating environmental impacts into health economic evaluation

The health sector is responsible for approximately five percent of global greenhouse gas (GHG) emissions, the majority of which arises from the lifecycle use of pharmaceuticals and medical devices[54]. These emissions have direct and indirect impacts on human health, health systems, and our planet through increased air, water, and environmental pollution, as well as frequent extreme weather events such as floods, droughts, and heatwaves[55]. Traditional health economic evaluations focus narrowly on direct expenditures and health outcomes, often overlooking these externalities. When such externalities are not reflected in the price, the health industry will continue to produce at the socially sub-optimal level. Consequently, decisions deemed cost-effective in the short term may inadvertently impose broader societal and economic burdens[56]. Therefore, there is growing call from the HTA community to incorporate the environmental impacts of health technologies into health economic evaluations[57,58].

Several HTA agencies including the Scottish Health Technologies Group, the National Institute for Health and Care Excellence (NICE), the CDA, the Institut national d'excellence en santé et en services sociaux(-INESSS), Zorginstituut Netherlands, and HITAP have begun exploring the integration of environmental impacts into healthcare decision-making[58]. In response, various methodological approaches have been proposed to support this effort. These include monetizing or converting environmental impacts into DALYs for inclusion in health economic evaluations; calculating an incremental carbon footprint-effectiveness ratio or incremental carbon footprint cost ratio; incorporating environmental impacts into multi-criteria decision analysis; and allowing for their deliberation during HTA processes[59].

However, adopting these methods presents new challenges. Estimates of environmental impacts are typically derived from Life Cycle Assessment (LCA) studies, which are primarily conducted within environmental and engineering disciplines. Consequently, essential data, such as emission factors required to calculate carbon footprints, are often lacking for health technologies[60]. Moreover, variations in data sources, assumptions, and system boundaries can lead to inconsistent results, complicating cross-study comparisons[61,62]. The recent launch of Lancet MedZero, a platform designed to provide robust and transparent carbon footprint data for all products used within health systems, has the potential to address some of these challenges[63]. To ensure that LCA findings are robust and suitable for integration into health economic evaluations, there is an urgent need for consensus and guidance on the conduct and reporting of healthcare LCA studies. Equally important is building the capacity of researchers to conduct health LCA studies, and of stakeholders to interpret and use such evidence in decision-making.

Lastly, some healthcare decisions offer clear co-benefits while others involve complex trade-offs. For example, a case study from an Australian teaching hospital found that switching from desflurane to sevoflurane during surgery reduced financial costs by approximately A\$14,630 while also decreasing GHG emissions by an amount equivalent to an average petrol car driving 1.4 million kilometers[64]. Conversely, other studies have highlighted trade-offs between climate and environmental sustainability goals and other sustainable development goals such as economic growth, equity, food security[65,66]. Although latter examples are not specific to health technologies, it is plausible that a health intervention could demonstrate superior safety and efficacy while imposing greater environmental harm, or conversely, be environmentally sustainable but economically costly. To navigate these challenges and align healthcare decisions with broader planetary and societal goals, it is essential to

establish a transparent and multisectoral framework that can support the systematic evaluation of such trade-offs. Understanding stakeholder preferences for such trade-offs may provide a valuable starting point for establishing such a framework[67,68].

### Evaluating innovative health technologies: New challenges beyond conventional frameworks

The rapid emergence of innovative health technologies, such as precision medicine and AI-driven tools, poses fundamental challenges to conventional health economic evaluation. These interventions often involve high upfront costs, dynamic performance, complex clinical pathways, and outcomes dependent on specific patient subgroups or real-world data. Traditional frameworks, designed for stable technologies and homogeneous populations, struggle to capture their full value, leading to significant uncertainty in pricing and reimbursement decisions[69,70]. Precision medicine creates value in healthcare by enabling more accurate diagnosis, personalized treatment, risk prediction, and improved clinical decision-making across the life course[71–74]. Its value extends beyond health outcomes to include benefits for families, health system efficiency, research innovation, and long-term reuse of genomic data that continues to generate clinical and societal gains over time. The multidimensional benefits of innovative technologies pose challenges for traditional health economic evaluation frameworks, which primarily focus on health outcomes and may therefore undervalue these technologies. These challenges are further compounded by evidence gaps, including limited long-term data, small sample sizes, and a lack of standardized outcome measures. Additional complexity arises from unclear target populations and the rapidly evolving nature of technologies. Advanced modelling approaches, such as microsimulation, are often required to capture complex test-treatment pathways. Finally, real-world implementation is constrained by challenges related to adoption and evolving healthcare systems.

To address these gaps, a more adaptive and continuous evaluation approach is essential. This includes proactive strategies such as early health economic assessment during development and adaptive pathways and managed entry agreements, which allow for iterative evidence generation. For example, experts within the HTAsiaLink network have developed specific guidelines to meet this need, creating the "PICCOTEAM" reference case for precision medicine[75]. New tools and methods should also be developed to capture the broader benefits of innovative health technologies, such as genomic testing, beyond direct health outcomes by incorporating patient-centred utility that reflects the perspectives of patients and their families[76–78]. Post-reimbursement, robust implementation research using real-world data remains critical for monitoring use and enabling evidence-based reassessment. Furthermore, advanced modeling techniques that integrate real-world data and multi-criteria decision analysis, combined with dynamic pricing, help align financial mechanisms with evidence development. Together, these strategies form a comprehensive framework for managing uncertainty, fostering sustainable innovation, and ensuring fiscal accountability.

### Conclusions

Health economic evaluation is a vital tool for promoting efficient and evidence-informed health policy which supports critical decisions across the technology lifecycle, from guiding innovation to informing coverage and disinvestment. However, its full integration requires addressing key challenges: moving beyond efficiency to incorporate equity, accounting for environmental costs, and adapting frameworks for novel technologies such as gene therapies and AI. Strengthening deliberative processes, building institutional capacity, and adopting more flexible methodologies are essential to harness its potential for building sustainable and equitable health systems.

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

## Acknowledgements

This research has received funding support from the Thailand Science Research and Innovation (TSRI), contract number FFB690031/0401. Hugo C Turner acknowledges funding from the MRC Centre for Global Infectious Disease Analysis (reference MR/X020258/1), funded by the UK Medical Research Council (MRC). This UK funded award is carried out in the frame of the Global Health EDCTP3 Joint Undertaking. The team acknowledges the input from Valentina Ricci from Agency for Care Effectiveness.

## Author contributions

Y.T. conceptualized the paper with all the authors, Y.T., Y.W., S.K., H.C.T., B.S.O., and W.I., wrote the first draft of the paper. All authors, Y.T., Y.W., S.K., H.C.T., B.S.O., and W.I. provided critical input, contributed to manuscript revisions, and approved the final submitted version.

## Competing interests

The authors declare no competing interests
