## [Transparent Peer Review file · Communications Medicine]

Harnessing health economic evaluation for policy and practice

Corresponding Author: Dr Yi WANG

Version 0:

Reviewer comments:

Reviewer #1

(Remarks to the Author)

This is a usual paper providing a nice overview of the potential of health economic evaluations to guide policy and decision making in healthcare.

Yet, I have some major and minor comments.

Major:

1. I miss the possibility for threshold modifiers. Admittedly, extended C-Eff is discussed, but a more widely and increasingly applied approach is to adapt CET in function of disease severity; this should be extensively discussed
2. I miss the role of budget impact in decision making as well. How to solve the cost-effective but unaffordable paradox. See for instance <https://www.tandfonline.com/doi/full/10.1080/13696998.2019.1632203>
3. Regarding Managed Entry Agreements, a distinction should be made between Outcomes Based and Financial Based MEAs.

Minor:

1. page 3, line 40: better call this here value for money, so that you better explain the difference between added therapeutical value (not yet taking into account the price) and value for money. See also page 4, line 3: should be value for money
2. sometimes you refer to economic evaluation, sometimes to health economic evaluation; please use the latter systematically
3. page 5, line 128: ICER was already discussed before, here it is presented as it is the first time mentioned in the paper...
4. page 6, line 161, please explain ACE in full: Singapore's national health technology assessment (HTA) and clinical guidance agency

Reviewer #2

(Remarks to the Author)

Overall comments

This paper positions economic evaluation as a core instrument throughout the health technology lifecycle from early HTA shaping product design to reimbursement, guideline development, and disinvestment. Asia-Pacific case studies demonstrate how cost-effectiveness evidence guides price negotiation, benefit package design, and removal of low-value care, while highlighting political barriers to evidence-based decisions. The study would benefit with greater detail and specificity of the insights presented.

Specific comments

1. The introductory paragraph to the "The role of economic evaluation across the healthcare technology lifecycle" feels like it is repeating the text from the last paragraph of the introduction. I suggest rewording the sentence "From assessing emerging innovations to guiding coverage decisions, pricing, guideline development, and disinvestment, health economic evaluations provide critical insights that support efficient and sustainable healthcare delivery" to something more actionable such as "health economic evaluations provide critical insights such X and Y for assessing emerging innovations to guiding coverage decisions, as well as Y and Z that inform pricing, guideline development, and disinvestment, therefore supporting efficient and sustainable healthcare delivery"
2. In Box 1 under the same section, it is not clear what comparators (if any) were used and what the criteria for cost-effectiveness was (e.g., potential for more diagnosis or lower costs compared to current alternatives). Please specify this information if available.

3. Although the name of the second section reads "Reimbursement decisions and price negotiation", the section does not actually contain any insights on price negotiation. Furthermore, the second paragraph presents the role of the ICER and cost-effectiveness thresholds in decision making. However, the definition of these terms is very well established in the health economic literature, and their inclusion in the second paragraph does not provide actionable insights on their relevance to decision making. I would suggest shortening the definition of the concepts and instead use the second paragraph to discuss how ICERs and CETs shape price negotiations and/or provide specific examples of how they guide reimbursement decisions in the two contexts presented (Australia and Philippines).

4. In section 4 "Other applications", please contextualise the use of surrogate outcomes in economic evaluations. Adding a one (or a few) specific examples of how surrogate outcomes have informed decisions in the APAC region can provide more value to the section.

5. Context is also missing in the "desflurane to sevoflurane" comparison in the section discussing environmental impacts. Please specify the context (e.g., surgery or anaesthesia) and setting (e.g., Australia)

6. Additional challenges to genetic technologies (and potentially precision medicine) are the multi-dimensional benefits achieved by genetic and genomic tests that are not captured in conventional frameworks such as QALYs. These challenges are very relevant to these interventions and fall outside the "operational efficiency" benefits described in the "Evaluating innovative health technologies section". The authors might want to consult relevant literature and incorporate those insights in this section: <https://doi.org/10.1016/j.gim.2024.101146> and <https://doi.org/10.1038/s41591-025-04061-3>

Version 1:

Reviewer comments:

Reviewer #1

(Remarks to the Author)

The authors have very well responded to my comments. No further remarks. Excellent work!

Reviewer #2

(Remarks to the Author)

Thank you for the revisions. All my comments have been addressed and I have no further recommendations to make. I do note that some of the sentences in the new text are lengthy, but I am sure these can be easily edited during copywriting.

Harnessing health economic evaluation for policy and practice (COMMSMED-25-2884)

Dear Editor and reviewers:

Thank you for the opportunity to revise our manuscript and for the insightful comments. We have revised the manuscript accordingly. Our point-by-point responses are provided below.

Reviewer 1:

This is a usual paper providing a nice overview of the potential of health economic evaluations to guide policy and decision making in healthcare. Yet, I have some major and minor comments.

Major:

1. I miss the possibility for threshold modifiers. Admittedly, extended C-Eff is discussed, but a more widely and increasingly applied approach is to adapt CET in function of disease severity; this should be extensively discussed

Response: We thank the reviewer for this insightful observation. We agree that threshold modification based on disease severity is an important and increasingly applied approach that merits discussion in our paper.

In response, we have added a paragraph at the end of the "Efficiency or equity or both? Operationalizing equity-informed analysis" section that addresses this point. The added paragraph discusses how several HTA jurisdictions, including the Netherlands and the United Kingdom, have operationalised severity-based threshold adjustments. We also incorporated lessons from Thailand, specifically the Imiglucerase case for Gaucher disease, to illustrate both the rationale for and challenges of such approaches. We highlighted the critical importance of grounding threshold modifications in domestic evidence on societal preferences and opportunity costs, and of implementing them alongside complementary policies such as managed entry agreements and competition-enhancing measures.

We believe this addition appropriately addresses the reviewer's comment and strengthens the paper's discussion of how equity considerations can be operationalised within cost-effectiveness frameworks.

The added text is presented below:

"Beyond distributional analyses, an increasingly adopted approach involves modifying CETs based on disease severity, reflecting societal preferences to prioritise those with greater illness burden. (42) The Netherlands applies higher thresholds (up to €80,000 per QALY) based on proportional disease burden, while the UK applies elevated thresholds for end-of-life and highly specialised technologies. (55, 56) However, implementing severity-based adjustments presents significant challenges, including varying definitions of severity, limited empirical basis for threshold magnitudes, and potential opportunity costs of redirecting resources from broader population health gains. (56) Thailand's experience with Imiglucerase for Gaucher disease illustrates these tensions: despite an ICER exceeding nine times the prevailing

threshold, the drug was approved for approximately five children annually. (25) Applying the value attribution framework developed by Ochalek et al (57), this decision resulted in negative realised population net health effects—meaning health gains for treated children were outweighed by health losses elsewhere in the system due to opportunity costs. Notably, subsequent market entry of a therapeutic competitor reduced Imiglucerase's price by over 25%, demonstrating that fostering competition, rather than simply raising thresholds, can improve both access and population health. Crucially, evidence from Thailand confirms that raising thresholds without corresponding price regulation increases manufacturer share without improving population health, underscoring that severity modifications should be grounded in domestic evidence on both societal preferences and opportunity costs, and implemented alongside complementary policies such as managed entry agreements and competition-enhancing measures. (57)”

2. I miss the role of budget impact in decision making as well. How to solve the cost-effective but unaffordable paradox. See for instance

<https://www.tandfonline.com/doi/full/10.1080/13696998.2019.1632203>

Response: We thank the reviewer for this important observation. The “cost-effective but unaffordable” paradox is indeed a central challenge for decision-makers, and we have substantially expanded the manuscript to address this issue, drawing on both conceptual frameworks and country experiences, including a new case study from Thailand.

In the section “Reimbursement decisions and price negotiation”, we have added a discussion of how jurisdictions manage this tension in practice. In particular, we now highlight that health economic evaluations are routinely considered alongside budget impact analyses to support more informed decisions on public reimbursement. To directly address the reviewer’s concern, we have incorporated the following text into the paragraph describing how countries such as Thailand navigate this paradox.

“A persistent challenge for health systems is the "cost-effective but unaffordable" paradox: an intervention may fall below a cost-effectiveness threshold yet still generate a budget impact that threatens fiscal sustainability. Countries have adopted various strategies to manage this tension, such as staggered implementation for hepatitis C treatment (27) and managed entry agreements for high-cost medicines like imiglucerase (28), which cap volumes or share financial risk with manufacturers. However, a core principle of health technology assessment is often misunderstood. HTA does not imply that all cost-effective interventions must be adopted. Rather, it is a priority-setting tool that enables decision-makers to say "no" to technologies that do not represent good value for money or are unaffordable, thereby safeguarding financial sustainability and avoiding the crowding out of other priority services. Evidence from Thailand illustrates this clearly: nearly one-third of proposed cancer medicines were never formally submitted for reimbursement because they were anticipated to be unaffordable, reflecting a form of pre-emptive rationing based on expected budget impact. (29) This highlights that affordability considerations may, at times, need to override cost-

effectiveness alone, and that transparent criteria and explicit trade-off decisions are essential to ensure limited resources maximise population health gains.”

3. Regarding Managed Entry Agreements, a distinction should be made between Outcomes Based and Financial Based MEAs.

Response: We have added the following under the section “Reimbursement decision and price negotiation”.

“Managed entry agreements can generally be classified into two types: financial-based and outcome-based. (22, 23) Financial-based agreements improve patient access while controlling costs by limiting reimbursed volume or reimbursement levels and sharing expenditures with manufacturers. Outcome-based agreements adjust and share payments according to the real-world performance of the therapies. Financial-based managed entry agreement has been used more often in Australia. (24)”

Minor:

1. page 3, line 40: better call this here value for money, so that you better explain the difference between added therapeutical value (not yet taking into account the price) and value for money. See also page 4, line 3: should be value for money

Response: Thanks for the comments. We have revised both terms as value for money,

2. sometimes you refer to economic evaluation, sometimes to health economic evaluation; please use the latter systematically

Response: We have revised the corresponding term as “health economic evaluation” throughout the manuscript.

3. page 5, line 128: ICER was already discussed before, here it is presented as it is the first time mentioned in the paper...

Response: We have revised the correspond sentence to “ICER, which compares the additional costs and health benefits of a new intervention relative to a comparator, together with the CET, provides the basis for assessing value for money” to serve as the starting point for the paragraph on decision making using ICER and CET.

4. page 6, line 161, please explain ACE in full: Singapore's national health technology assessment (HTA) and clinical guidance agency

Response: the sentence has been revised as the following “Box 2 presents an example of this application by Agency for Care Effectiveness (ACE), Singapore's national HTA and clinical guidance agency.”

We thank reviewer’s comments which have significantly improved our manuscript.

Reviewer 2:

Overall comments

This paper positions economic evaluation as a core instrument throughout the health technology lifecycle from early HTA shaping product design to reimbursement, guideline development, and disinvestment. Asia-Pacific case studies demonstrate how cost-effectiveness evidence guides price negotiation, benefit package design, and removal of low-value care, while highlighting political barriers to evidence-based decisions. The study would benefit with greater detail and specificity of the insights presented.

Response: We thank the reviewer for the helpful comments. We have provided additional details and revised the manuscript accordingly.

Specific comments

1. The introductory paragraph to the "The role of economic evaluation across the healthcare technology lifecycle " feels like it is repeating the text from the last paragraph of the introduction. I suggest rewording the sentence "From assessing emerging innovations to guiding coverage decisions, pricing, guideline development, and disinvestment, health economic evaluations provide critical insights that support efficient and sustainable healthcare delivery" to something more actionable such as "health economic evaluations provide critical insights such X and Y for assessing emerging innovations to guiding coverage decisions, as well as Y and Z that inform pricing, guideline development, and disinvestment, therefore supporting efficient and sustainable healthcare delivery"

Response: Thank you for the suggestion. The sentence has been revised as the following "Health economic evaluations provide critical insights, such as comparative effectiveness and value for money to assess emerging innovations, as well as evidence on cost-effectiveness and budget impact to inform pricing, guideline development, and reevaluation decisions that may guide disinvestment, thereby supporting efficient and sustainable healthcare delivery."

2. In Box 1 under the same section, it is not clear what comparators (if any) were used and what the criteria for cost-effectiveness was (e.g., potential for more diagnosis or lower costs compared to current alternatives). Please specify this information if available.

Response: Thank you for the comments. The following information has been added to Box 1: "Tongue swab with real-time polymerase chain reaction (RT-PCR) and tongue swab with Loop-Mediated Isothermal Amplification (LAMP) were compared with the current practices, including acid-fast bacillus smear microscopy with sputum Xpert testing for individuals aged above 5 years and tuberculin skin test for children under age 5 years. Using the Thai CET 160,000 Thai Baht per QALY (17), tongue swab with RT-PCR were found to be cost-effective for individuals age above 5, but tongue swab with LAMP was not. For children under 5 years, both tongue swab with RT-PCR and tongue swab with LAMP were cost-effective."

3. Although the name of the second section reads "Reimbursement decisions and price negotiation", the section does not actually contain any insights on price negotiation.

Furthermore, the second paragraph presents the role of the ICER and cost-effectiveness thresholds in decision making. However, the definition of these terms is very well established in the health economic literature, and their inclusion in the second paragraph does not provide actionable insights on their relevance to decision making. I would suggest shortening the definition of the concepts and instead use the second paragraph to discuss how ICERs and CETs shape price negotiations and/or provide specific examples of how they guide reimbursement decisions in the two contexts presented (Australia and Philippines).

Response: We thank the reviewer for this constructive feedback. We agree that the section would benefit from more actionable insights on price negotiation and a shortened definition of ICERs and CETs. We have substantially revised this section as follows:

- Removed the detailed definition of ICERs and CETs (now briefly noted in one sentence)
- Replaced with a new paragraph synthesising how different Asian countries use CETs in price negotiations—incorporating the reviewer's suggested text while integrating it more smoothly with the existing content
- Retained the core examples (Australia, Philippines) but repositioned them within this comparative framework
- Added in additional discussion on the cost-effective but unaffordable paradox.

As the entire section has been revised, the reviewer can refer to the revised manuscript for the updated text.

4. In section 4 "Other applications", please contextualise the use of surrogate outcomes in economic evaluations. Adding a one (or a few) specific examples of how surrogate outcomes have informed decisions in the APAC region can provide more value to the section.

Response: We thank the reviewer for this suggestion and have revised the section to include more details including adding a recently published manuscript (Bujkiewicz S, Ciani O, Heeg B, Lee D, Kusel JM, Thorlund K, Pechlivanoglou P, Stefani S, Isaranuwatthai W, Buyse M, Ouwens M. Methods for Evaluation of Surrogate Endpoints for HTA Decision Making: A Good Practices Report of an ISPOR Task Force. Value in Health. 2026 Feb 10.)

The following sentences were added: "A Good Practices Report on methods for evaluation of surrogate endpoints for HTA decision making emphasizes validating surrogates at three levels: 1) treatment effect association (does treatment change in the surrogate predict change in the final outcome?); 2) outcome association (are surrogate and final outcome associated?); and 3) biological plausibility. Recommended methods include multivariate meta-analysis, which accounts for differences between studies and correlations within them; Bayesian approaches, which help strengthen conclusions when data are limited; and the use of real-world data, which can complement trial evidence, provided that potential biases are carefully assessed and managed. (40) An example could be when an individual participant data (IPD) meta-analysis was conducted to assess surrogacy, showing that disease-free survival is a strong surrogate for overall survival in HER2-positive breast cancer patients treated with trastuzumab. (41) However, when Canada's Drug Agency assessed pertuzumab with trastuzumab and

taxane for neoadjuvant treatment of HER2-positive breast cancer, an IPD meta-analysis found that pathological complete response did not reliably predict overall survival. (42) Due to this weak association, the drug was not recommended."

5. Context is also missing in the "desflurane to sevoflurane" comparison in the section discussing environmental impacts. Please specify the context (e.g., surgery or anaesthesia) and setting (e.g., Australia)

Response: Thank you for the suggestion. We have now added text to provide more context around the aesthetic gas example. The revised text is as follows:

"Lastly, some healthcare decisions offer clear co-benefits while others involve complex trade-offs. For example, a case study from Australian teaching hospital found that switching from desflurane to sevoflurane during surgery reduced financial costs by approximately A\$14,630 while also decreasing GHG emissions by an amount equivalent to an average petrol car driving 1.4 million kilometers. (68) Conversely, other studies have highlighted trade-offs between climate and environmental sustainability goals and other sustainable development goals such as economic growth, equity, food security. (69, 70) Although latter examples are not specific to health technologies, it is plausible that a health intervention could demonstrate superior safety and efficacy while imposing greater environmental harm, or conversely, be environmentally sustainable but economically costly. To navigate these challenges and align healthcare decisions with broader planetary and societal goals, it is essential to establish a transparent and multisectoral framework that can support the systematic evaluation of such trade-offs. Understanding stakeholder preferences for such trade-offs may provide a valuable starting point for establishing such a framework. (71, 72)"

We have also taken this opportunity to update this section with recent developments and discussion in the filed including the launch of the Lancet MedZero, which aims to provide carbon footprint data for all products used by health systems – addressing some of the challenges highlighted in the manuscript. The following text has been added "The recent launch of Lancet MedZero – a platform designed to provide robust and transparent carbon footprint data for all products used within health systems – has the potential to address some of these challenges. (67)"

6. Additional challenges to genetic technologies (and potentially precision medicine) are the multi-dimensional benefits achieved by genetic and genomic tests that are not captured in conventional frameworks such as QALYs. These challenges are very relevant to these interventions and fall outside the "operational efficiency" benefits described in the "Evaluating innovative health technologies section". The authors might want to consult relevant literature and incorporate those insights in this section: <https://doi.org/10.1016/j.gim.2024.101146> and <https://doi.org/10.1038/s41591-025-04061-3>

Response: Thanks for your suggestions and the additional references. The section has been expanded to better reflect the current literature.

“The rapid emergence of innovative health technologies, such as precision medicine and AI-driven tools, poses a fundamental challenge to conventional health economic evaluation. These interventions often involve high upfront costs, dynamic performance, complex clinical pathways, and outcomes dependent on specific patient subgroups or real-world data. Traditional frameworks, designed for stable technologies and homogeneous populations, struggle to capture their full value, leading to significant uncertainty in pricing and reimbursement decisions. (73, 74) Precision medicine creates value in healthcare by enabling more accurate diagnosis, personalized treatment, risk prediction, and improved clinical decision-making across the life course. (75-78) Its value extends beyond health outcomes to include benefits for families, health system efficiency, research innovation, and long-term reuse of genomic data that continues to generate clinical and societal gains over time. The multidimensional benefits of innovative technologies pose challenges for traditional health economic evaluation frameworks, which primarily focus on health outcomes and may therefore undervalue these technologies. These challenges are further compounded by evidence gaps, including limited long-term data, small sample sizes, lack of standardized outcome measures; complex use cases arising from unclear target populations and the rapidly evolving nature of technologies; the need for advanced modelling approaches such as microsimulation due to complex test-treatment pathways; and real-world implementation constraints related to adoption and changing healthcare systems.”

“New tools and methods should also be developed to capture the broader benefits of innovative health technologies, such as genomic testing, beyond direct health outcomes by incorporating patient-centred utility that reflects the perspectives of patients and their families. (80-82)”

We thank reviewer’s comments which have significantly improved our manuscript.